# Lipid Metabolism in Glioblastoma: From De Novo Synthesis to Storage

**DOI:** 10.3390/biomedicines10081943

**Published:** 2022-08-11

**Authors:** Yongjun Kou, Feng Geng, Deliang Guo

**Affiliations:** 1Department of Radiation Oncology, Ohio State Comprehensive Cancer Center, Arthur G. James Cancer Hospital and Richard J. Solove Research Institute, College of Medicine at The Ohio State University, Columbus, OH 43012, USA; 2Center for Cancer Metabolism, James Comprehensive Cancer Center at The Ohio State University, Columbus, OH 43210, USA

**Keywords:** glioblastoma, fatty acids, cholesterol, lipid droplets, SREBP-1, DGAT1, SOAT1, lipotoxicity

## Abstract

Glioblastoma (GBM) is the most lethal primary brain tumor. With limited therapeutic options, novel therapies are desperately needed. Recent studies have shown that GBM acquires large amounts of lipids for rapid growth through activation of sterol regulatory element-binding protein 1 (SREBP-1), a master transcription factor that regulates fatty acid and cholesterol synthesis, and cholesterol uptake. Interestingly, GBM cells divert substantial quantities of lipids into lipid droplets (LDs), a specific storage organelle for neutral lipids, to prevent lipotoxicity by increasing the expression of diacylglycerol acyltransferase 1 (DGAT1) and sterol-O-acyltransferase 1 (SOAT1), which convert excess fatty acids and cholesterol to triacylglycerol and cholesteryl esters, respectively. In this review, we will summarize recent progress on our understanding of lipid metabolism regulation in GBM to promote tumor growth and discuss novel strategies to specifically induce lipotoxicity to tumor cells through disrupting lipid storage, a promising new avenue for treating GBM.

## 1. Introduction

Fatty acids and cholesterol are two essential lipids that constitute the basic structure of cellular membranes [1]. Their derivatives can act as important signaling molecules to regulate cellular function [2]. However, dysregulation of fatty acid and cholesterol metabolism can cause various metabolic diseases, i.e., obesity, fatty liver, atherosclerosis, neutral lipid storage disease (NLSD), and cholesterol storage disease (CESD) [3,4]. Increasing evidence shows that lipid alterations also accelerate neurodegenerative disease progression by promoting oligomerization of pathogenic proteins and brain inflammation [5,6]. In recent years, following extensive investigation on energy metabolism [7], the role of abnormal lipid metabolism in cancer cells has rapidly received increasing attention [1,8,9,10]. New observations indicate addictive de novo lipid synthesis in various types of cancers [1]. Thus, targeting lipid synthesis has become a promising new direction for cancer therapy, as demonstrated by the many preclinical tumor models and Phase I/II clinical trials (Table 1) [1,9].

Glioblastoma (GBM) is the most lethal primary brain tumor [11]. Unfortunately, there has not been significant progress to improve GBM outcome over the past two decades despite extensive clinical testing [12]. Recent studies show that GBM needs abundant lipids for rapid growth [13,14,15,16], but accumulation of intracellular lipids, especially free fatty acids and cholesterol, can cause lipotoxicity [17,18]. How GBM cells protect themselves against excess lipid accumulation and lipotoxicity is an intriguing question. Our recent studies demonstrated that GBM cells prevent lipotoxicity and maintain tumor growth by dynamically storing excess lipids into lipid droplets (LDs), a specific lipid storage organelle [19,20,21,22]. In this review, we will summarize recent progress on our understanding of lipid metabolism regulation in GBM to maintain cellular homeostasis and tumor growth via coordination of lipid synthesis, uptake and storage processes, and will discuss a new strategy to target GBM involving disruption of lipid storage specifically in tumor cells.

## 2. Standard GBM Therapy

GBM is a WHO grade IV glioma, with a median survival post-diagnosis of only approximately 15 months despite extensive treatments [23]. GBM represents more than 60% of all brain tumors in adults and occurs in 2–3 cases per 100,000 individuals in Europe and North America [24]. Approximately 90% of GBM cases occur as primary tumors that have wild-type isocitrate dehydrogenase (IDH) and less than three months of clinical symptoms before diagnosis, while less than 5% of all cases are secondary tumors with IDH mutations, progressing from WHO grade II/III gliomas and with better prognosis [12,25,26,27]. The standard treatment for GBM is to first perform maximal surgical resection, followed by concomitant radiotherapy and alkylating agent Temozolomide (TMZ) for 6 weeks, and then continuation of TMZ alone every 4 weeks for six cycles of maintenance treatment [28]. Unfortunately, this treatment is only effective for a few months and almost all GBM tumors unavoidably recur after treatment [29]. The 5-year survival for patients with GBM is only around 6.8% in the United States [30].

In the clinic, except for TMZ, only Bevacizumab, a humanized monoclonal antibody against vascular endothelial growth factor (VEGF), has been approved by the FDA for the treatment of recurrent GBM and not for newly diagnosed GBM [31,32]. It is effective at reducing symptoms and improving the quality of life only for a short time [25]. However, Bevacizumab has no combinatory effect with TMZ and radiotherapy [25]. Moreover, it reduces the uptake of TMZ, fails to increase the overall survival, and has significant side effects such as hypertension, venous thrombosis, and infections [25].

Current investigations for new approaches targeting GBM have centered on molecularly targeted therapies, immunotherapy, and brain-targeted drug delivery [33]. For example, 57% of GBM cases show evidence of amplification or gain of function mutation of EGFR (EGFRvIII) [34]. Unfortunately, results from clinical trials for monoclonal antibodies and tyrosine kinase inhibitors (TKI) targeting EGFR have been disappointing, with no therapeutic benefit or being no better than TMZ [12,35]. Moreover, although 61% [36] to 88% [37] of GBM patients have tumors expressing PD-L1, clinical trials inhibiting PD-L1 or PD1 have also shown disappointing results [38]. Collectively, the failure to make significant progress in targeting GBM indicates that more efforts are needed to understand the biology of GBM, which is foundational to identify effective approaches antagonizing GBM.

## 3. Lipid Metabolism Regulation in GBM

### 3.1. Lipogenesis

Glucose is the main source for lipid production through the de novo synthesis pathway [1]. Glucose through glycolysis generates pyruvate that is converted to acetyl-CoA in the mitochondria by the pyruvate dehydrogenase (PDH) [1,39]. Condensation of acetyl-CoA with oxaloacetate (OAA) forms citrate, which is then released into the cytosol by the SLC25A1 transporter and converted back to acetyl-CoA by ATP-citrate lyase (ACLY). Acetyl-CoA then serves as a precursor for fatty acid and cholesterol synthesis [1,40] (Figure 1). In addition, acetyl-CoA synthetase 2 (ACSS2) can convert cytosolic acetate to acetyl-CoA for de novo fatty acid and cholesterol synthesis (Figure 1) to promote tumor growth [1]. ACSS2 is upregulated in GBM [41] and in hepatocellular carcinoma (HCC), myeloma, prostate, and bladder cancers [42]. Furthermore, acetyl-CoA is also involved in epigenetic regulation as it enters into the nucleus to modify histone proteins by direct acetylation [43].

De novo fatty acid synthesis is commonly observed in the liver, adipose tissue, and lactating breast, and normal cells usually assimilate fatty acid through the uptake of extracellular lipids [44]. However, actively proliferating cells, such as cancer cells and effector T cells, increase their fatty acid synthesis, uptake, and oxidation (FAO) for the synthesis of structural lipids and energy expenditure [9,45]. Expression of the enzymes controlling de novo fatty acid synthesis, such as ACLY, ACSS2, acetyl-CoA carboxylases (ACC), fatty acid synthase (FASN), and stearoyl-CoA desaturase 1 (SCD1), and mitochondria citrate transporter SLC25A1, is mainly regulated by sterol regulatory element-binding protein 1 (SREBP-1), a master transcription factor that regulates fatty acid and cholesterol synthesis (Figure 1) [13,14,15,46,47]. Inhibitors targeting these enzymes have been developed for the treatment of a broad type of cancers (Table 1) [9]. Carnitine palmitoyltransferase 1 (CPT1), the rate-limiting enzyme involved in mitochondrial fatty acid oxidation, which allows fatty acyl-CoA to enter into the mitochondria matrix from the cytosol, is also an important target of lipid metabolism for cancer therapy [48,49].

Like in most cancers, metabolism is reprogrammed in GBM and plays an important role during tumor initiation and progression, with increased glycolysis, altered nucleotide metabolism, and glutamine addiction [50,51]. We were the first to report that fatty acid synthesis is highly elevated in GBM to promote rapid tumor growth [1,13,14,15]. We found that the oncogenic EGFR/PI3K/AKT pathway increases fatty acid synthesis by stimulating the activation of SREBP-1 [14,39]. Inactivating SREBP-1 pharmacologically with its inhibitor Fatostatin or by shRNA knockdown significantly suppresses GBM tumor growth in vitro and in vivo [14,19,52]. Directly targeting FASN, a downstream transcriptional target of SREBP-1 and a key enzyme in controlling de novo fatty acid synthesis, with its inhibitor C75 effectively suppresses GBM growth [14]. Moreover, inhibiting fatty acid and cholesterol synthesis with the AMP-activated protein kinase (AMPK) agonist AICAR (5-aminoimidazole-4-carboxamide-1-β-D-ribofuranoside) also reduces GBM growth in a xenograft mouse model [13,16]. Furthermore, our studies revealed that a small non-coding RNA, miR-29, mediates a negative feedback loop to control SREBP expression and lipogenesis [53,54]. SREBP-1 transcriptionally upregulates pre-miR-29 expression, while mature miR-29 can inversely inhibit SREBP-1 expression via binding to the 3′-untranslational region (3-UTR) of SREBP-1 mRNA to induce its degradation and suppress SREBP translation. Delivering miR-29 to GBM cells significantly suppresses tumorigenesis in orthotopic mouse model by inhibiting SREBP-1 and lipogenesis [53,54]. Collectively, these data demonstrate that inhibiting lipogenesis is a promising new direction for GBM therapy.

Interestingly, a recent study reports that the brain metastasis from seven breast cell lines that harbor PI3K activation is dependent on the SREBF1 gene that encodes the SREBP-1 protein [55]. A significant inhibition of metastatic brain is observed in SREBF1 knockout breast cancer cell lines [55], demonstrating the importance of SREBP-1 in brain metastasis from breast cancers. Whether other types of brain metastatic cell lines have a similar SREBF1 dependency needs further investigation. In addition, directly targeting SREBP-1 downstream targets should be tested in these cell lines to determine whether fatty acid or cholesterol synthesis play major roles for breast cancer brain metastasis.

### 3.2. SREBP Activation, Connecting Glucose and Glutamine to Lipid Synthesis

GBM growth consumes large amounts of glucose and amino acids [56], but the mechanism by which tumor cells sense their levels to trigger lipid synthesis has been left unanswered for a long time. Our most recent study identified that glucose and glutamine coordinate to activate SREBP-1 to trigger de novo lipid synthesis in GBM and various other cancer cells [57]. The SREBP family includes three isoforms, SREBP-1a, -1c, and -2 [58,59,60]. SREBP-1c mainly regulates the expression of genes controlling fatty acid synthesis, while SREBP-2 regulates cholesterol synthesis and uptake, and SREBP-1a, which has the highest transcriptional activity, regulates all three processes [1,61,62,63]. SREBPs are synthesized as ~125 kD inactive precursors, which are spatially restrained in the endoplasmic reticulum (ER) membrane and are activated through a tightly controlled ER–Golgi–nucleus translocation process [59,62]. SREBPs bind to SREBP-cleavage activating protein (SCAP), which further binds to COPII-coated vesicles to move from the ER to the Golgi [62,64]. In the Golgi, SREBPs are sequentially cleaved by site-1 and -2 proteases to release their N-terminal forms (~65 kD) that then enter the nucleus to activate lipogenic gene expression [65,66,67,68,69]. Interestingly, SCAP/SREBP trafficking is inhibited by an ER-resident protein, insulin-inducible gene protein (Insig), which includes two isoforms, Insig-1 and -2 [70,71]. Insig binds to SCAP to retain the SCAP/SREBP complex in the ER (Figure 1) [59,64]. Cholesterol or 25-hydroxycholesterol (25-HC) can bind to SCAP or Insig to further enhance their association, which represents a negative feedback loop to modulate SREBP activation (Figure 1) [70,72,73].

The key step activating the SCAP/Insig dissociation for subsequent SREBP translocation has just been elucidated in a recent study by Cheng et al. [57]. Our previous study showed that glucose stimulates SREBP activation and lipogenesis by promoting SCAP N-glycosylation and stability [46,74,75,76]. Unexpectedly, our most recent study shows that when glutamine is removed from the medium, glucose alone is unable to activate SREBPs and lipogenesis, even with low cholesterol levels and in the presence of SCAP N-glycosylation [57]. We uncovered an unprecedented role of ammonia, which is released by glutaminolysis and acts as a key activator of the dissociation of N-glycosylated SCAP from Insig by inducing dramatic conformational changes in the SCAP transmembrane domain through interaction via hydrogen bonds with the side chains of three residues, i.e., aspartate D428, serine S326 and S330, eventually leading to SREBP activation and lipid synthesis (Figure 1) [57]. We further unveiled the competitive role of 25-HC that blocks ammonia binding to SCAP, thereby keeping SCAP bound to Insig and suppressing SREBP activation. Blocking ammonia binding to SCAP by mutating D428 to alanine (D428A) dramatically suppressed both GBM and lung cancer growth in orthotopic xenograft models. In addition, we provided further physiological evidence for a connection between glutaminolysis and lipogenesis by showing the molecular link between glutaminase (GLS) expression and SREBP-1 activation in human lung cancer and glioma tissues. Moreover, the study demonstrated that the activation of SREBPs and lipogenesis by glutamine/ammonia in concert with glucose also occurs in melanoma, liver and breast cancer cells in addition to lung cancer and GBM cells, suggesting that this is a common mechanism at play in a wide range of cancer types [57]. Thus, this study revealed the essential role of ammonia in the regulation of SCAP/Insig dissociation, SREBP activation and lipid metabolism, and identified SCAP as a critical sensor connecting glutamine, glucose, and lipid metabolism to promote tumor growth (Figure 1) [57]. The study also suggests that targeting the key molecular link between glutaminolysis, glucose and lipid metabolism might be a promising strategy for treating malignancies and metabolic syndromes.

### 3.3. Membrane Phospholipid Remodeling

Phospholipids (PLs) constitute the main membrane structure as they form lipid bilayers through their amphiphilic characteristics [77]. Elevated fatty acid synthesis promotes PL formation and membrane expansion to facilitate rapid tumor growth [1,78,79]. The composition of the fatty acid chains in PLs is not static, but is dynamically remodeled through the coordinated activities of phospholipase A (PLA) and lysophosphatidylcholine acyltransferases (LPCATs) [80]. PLA removes an esterified fatty acid from the sn-2 position of PLs to convert them to lysophosphatidylipids, while LPCATs add back fatty acids to lysophosphatidylipids, thereby remaking PLs [81,82]. Thus, fatty acid chains in membrane PLs are modified by these two enzymes to adjust to the dynamic alterations of the cellular environment [82]. The LPCAT family contains four members (LPCAT1-4) [83,84]. LPCAT1 mainly regulates the incorporation of saturated fatty acids into lysophosphatidylcholine (LPC) to form PLs, thereby increasing the levels of saturated PLs. Interestingly, LPCAT1 has been shown to be upregulated in various cancers, including GBM [85], HCC [86], clear cell renal cell cancer [87], esophageal squamous cell carcinoma (ESCC) [88], gastric [89], and breast cancer [90]. Its upregulation regulates oncogenic EGFR signaling in GBM by affecting EGFR and its constitutively active mutant EGFRvIII protein stability [85]. Targeting LPCAT1 effectively suppresses GBM [85] and ESCC growth [88]. LPCAT1 elevation across cancer types suggests that the increase in membrane saturated PL levels might prevent lipid peroxidation and ferroptosis in cancer cells, thereby facilitating tumor growth. LPCAT3 regulates the incorporation of polyunsaturated fatty acids, such as arachidonic acid, thereby increasing membrane fluidity and flexibility [84]. LPCAT3 has been shown to be a major player to facilitate lipid peroxidation and ferroptosis via increasing membrane polyunsaturated PL level [91,92,93,94]. Nevertheless, there has no study clearly demonstrating the expression levels of LPCPAT3 in tumor tissues and its role in tumorigenesis.

### 3.4. Cholesterol Uptake

Cholesterol, which is an indispensable lipid molecule for maintaining cell viability [95], is inserted between phospholipid molecules to control membrane integrity and fluidity [96], and accounts for 30–40% of membrane lipids in the plasma membrane [97,98,99,100]. Cholesterol is esterified with fatty acids to convert to neutral cholesteryl esters (CE), which are carried by low-density lipoprotein (LDL) to circulate in the bloodstream and be absorbed by different tissues [101]. LDL is endocytosed into cells as mediated by the LDL receptor (LDLR) and is then delivered to the lysosomes, where LDL is broken down by lysosomal lipases to release free cholesterol, which then egresses from the lysosomes for maintenance and expansion of the plasma membrane but also of the membranes of all cellular organelles such as the mitochondria and endoplasmic reticulum (ER) (Figure 1) [15,102,103,104,105,106].

Recent studies from our group showed that GBM cells acquire abundant cholesterol for rapid growth by elevating LDL uptake [15]. GBM increases the expression of the LDL receptor (LDLR), which enhances LDL absorption to provide sufficient cholesterol [15]. We further showed that EGFR signaling activates SREBP-1 to upregulate LDLR expression and promote cholesterol uptake for GBM growth [15]. Inducing LDLR degradation by increasing expression of its E3 ligase Idol (inducible degrader of LDLR) with the LXR agonist GW3965 inhibits GBM growth [15]. Therefore, limiting LDL/cholesterol uptake appears as a promising new direction for GBM therapy.

### 3.5. Lipid storage and Energy Homeostasis

Excess free fatty acids and cholesterol can cause lipotoxicity, leading to cell death when the levels are uncontrolled [107,108]. In the human body, excess fatty acids and cholesterol are sequestered into the adipocytes of fat tissues within specialized organelles known as lipid droplets (LDs) after being converted to triacylglycerol (TAG) and cholesteryl esters (CE) (Figure 1) [109]. Interestingly, in 2016, Geng et al. reported for the first time that LDs are highly prevalent in human GBM tumor tissues, while not observed in normal brain tissues [19,20]. Using transmission electronic microscopy, they clearly showed that LDs are present in the cytosol of GBM tumor cells. In following studies, the group demonstrated that when glucose level decreases, GBM cells mobilize LD-stored fatty acids into the mitochondria, a process mediated by autophagy for beta-oxidation and energy production, which maintain energy homeostasis and GBM survival [21]. Thus, LDs serve two functions in GBM. First, they act as a lipotoxicity-preventing organelle, buffering excess fatty acids and cholesterol [17,22]. Second, they also serve as an energy reservoir, providing fuel to maintain energy homeostasis when glucose levels are reduced [21]. Thus, targeting LD formation could be a promising new approach for GBM therapy, which is discussed in the sections below.

## 4. The Basics of LDs

### 4.1. LD Formation

LDs are organelles containing a core of neutral lipids, mainly TAG and CE, which is covered by a monolayer of phospholipids [4,109,110,111,112]. LDs exist in almost all organisms, from bacteria, yeast, and plants to mammals [113]. Mammalian LDs were first described in 1886 with improvement in microscopy technology but were ignored for more than a century [114]. In the 1990s, the discovery of perilipin, the first LD membrane protein to be identified, and that of Chanarin–Dorfman Syndrome, the first genetic LD disorder [114], incited scientists to begin exploring the characteristics of LDs and their roles in metabolic diseases. Nevertheless, despite major advances in recent years, the mechanism of LD formation is still unclear [109]. Three hypotheses have been advanced to explain LD formation, and the “budding model” is the most widely accepted [115]. The first step in LD biogenesis is the synthesis of TAG and CE derived from fatty acids and cholesterol in the ER, which are catalyzed by diacylglycerol acyltransferases (DGAT1 and DGAT2) and sterol O-acyltransferases (SOAT1 and SOAT2), also named acyl-CoA:cholesterol acyltransferases (ACAT1 and ACAT2) [109]. As the concentration of TAG and CE increases in the interspace between the bilayer leaflets of the ER membrane, the neutral lipids eventually bud from the ER membrane when the surface tension reaches a critical point and get covered with a phospholipid monolayer derived from the ER two leaflets [109,116,117,118]. This budding process is facilitated by multiple ER proteins, such as the fat-storage-inducing transmembrane (FIT) proteins and Seipin [109,119,120,121,122,123,124]. Because of the instability of the phospholipid monolayer, perilipin traffics from the ER to the surface of LDs to reduce the membrane surface tension. It is still unclear whether LDs bud from ER randomly or from preferential sites. Finally, LDs grow and mature through droplet-to-droplet fusion or lipid synthesis on the LD surface [109,124,125,126,127,128].

### 4.2. LD Size and Composition

The most predominant features of LDs are their heterogeneity and diversity. The number, size, and composition of LDs vary widely between cells or even within the same cell depending on different conditions [109,112,129,130,131]. Only one LD with a diameter often >100 µm is found in most adipocytes, whereas a greater number of LDs are found in non-adipocyte cells and are usually <1 µm in diameter [132,133]. However, the size and number of LDs change dramatically under pathophysiological conditions. For example, during the progression of non-alcoholic steatohepatitis (NASH), the increased number and size of LDs in hepatocytes causes the activation of hepatic stellate cells, leading to inflammation and liver fibrosis [134]. Interestingly, LDs are loaded with excess oleic acid and cholesterol in adrenal and liver cells, and two subpopulations of LDs, i.e., TG-rich and CE-rich LDs, are found in the cytosol and coated with different members of the perilipin family [135]. Translation inhibitors only induce formation of CE-rich LDs in 3Y1 cells by an unknown mechanism [136]. The preferential lipid composition of LDs and how it is established needs further investigation.

### 4.3. LD Interaction and Proteomes

LDs are cytoplasmic organelles that tightly communicate with other cellular organelles, such as the ER, mitochondria, peroxisomes, Golgi apparatus, and lysosomes [137]. LDs extensively form membrane contact sites to exchange materials such as lipids, metabolites, and ions with other organelles [109,138,139], as well as to regulate organelle division, trafficking, and inheritance [109,140,141]. Dysregulation of LD contact site proteins are found in many genetic, infectious, and metabolic diseases [137]. Recently, LDs were found to reside also in the cell nucleus and were termed “nuclear LDs (nLD)” [142,143,144,145]. Although the mechanism of nLD formation is still unclear, their function in gene transcription regulation has been described in yeast [146], hepatic cell lines [147], and primary hepatocytes [148].

Because of the extensive contact of LDs with other organelles, purification by optimized fractionation approaches still contain contaminants from other associated organelles, especially the ER [149], which has hampered the identification of the profile of LD proteomes. However, the recent development of the proximity labeling strategy enables high-confidence cataloguing of LD proteomes, which are composed of 100–150 proteins in mammalian cells [112,150]. These proteins are categorized into several distinct functional classes, involving structure maintenance, lipid metabolism, signaling, and membrane trafficking [4]. They can also be classified into two classes according to differences in trafficking pathways, i.e., class I LD proteins that are first inserted into the ER through hydrophobic hairpins and then traffic to the surface of LDs through membrane bridges, and class II LD proteins that are synthesized in the cytosol and inserted into LDs through a lipid anchor or amphipathic α-helices [109,112]. LD protein degradation is mediated by the ubiquitin-dependent pathway or the autophagic pathway [112].

## 5. Inhibiting LD Formation for GBM Therapy

LDs had been reported to be abundant in many cancer types, although not in every subtype [111]. In the early 1960s, LDs were found in the mammary carcinoma of a female patient [151] and abundantly in more than 100 cases of malignant lymphoma [152]. However, few people realized the importance of LDs in cancer research at that time. LDs have now been suggested to be involved in all processes of cancer development, including initiation, promotion, and progression [111]. The hypotheses regarding LD function in cancer cells are numerous, i.e., promoting tumor proliferation [153,154,155], driving epithelial–mesenchymal transition (EMT) [115,156,157], mediating radio-resistance [158,159] and chemo-resistance [160,161,162,163], maintaining cancer cell stemness [164,165,166], and avoiding immune destruction [167,168,169]. Nevertheless, these hypotheses need to be experimentally validated.

Our group reported for the first time that human patients with glioma exhibit LDs in large numbers that correlated with GBM progression and poor survival [19]. Our studies suggested that targeting SOAT1 and DGAT1 could be a promising new strategy for GBM therapy as neutral lipid synthesis catalyzed by SOAT1 and DGAT1 initiates the formation of LDs in GBM

### 5.1. Targeting SOAT1 to Disrupt Cholesterol Homeostasis for GBM Therapy

Cholesterol is an essential lipid, but excess cholesterol is toxic to cells; thus, the intracellular cholesterol levels need to be precisely balanced [1,20]. Cholesterol homeostasis is regulated through a complex network, including uptake, biosynthesis, efflux, storage, and trafficking [1,96,170]. Disrupting this network could trigger toxicity for cells. The ER is the key cellular organelle that senses changes in cholesterol levels. A previous study using Chinese hamster ovary (CHO) cells as a model showed that even a low 5% increase in ER cholesterol can trigger a feedback reaction that blocks SREBPs from trafficking to the Golgi and being activated, which slows down cholesterol synthesis and uptake in order to maintain cholesterol homeostasis [58,171]. Intracellular cholesterol levels can also be reduced by promoting cholesterol efflux through the ATP-binding cassette (ABC) transporter superfamily, including the ABC subfamily A 1 (ABCA1) and ABC subfamily G (ABCG) 1, 5, and 8 members in a cell-type dependent manner [96,172,173]. Our previous study showed that promoting cholesterol efflux and at the same time reducing cholesterol uptake by activating the liver X receptor (LXR) inhibits GBM growth by disturbing cholesterol balance [15].

Another strategy developed by cells to overcome lipotoxicity from excess cholesterol is to convert free cholesterol to CE in the ER and then store CE in LDs, a process controlled by SOATs [174,175]. SOATs belong to the membrane-bound *O*-acyltransferase (MBOAT) superfamily [176] and include two isoenzymes, i.e., SOAT1 and SOAT2. SOAT1 is mainly located in the ER and has been detected in most cell types and tissues, while SOAT2 is expressed predominantly in fetal liver and intestine cells and at low levels in other tissues [177,178,179]. SOATs, especially SOAT1, have long been studied as a therapeutic target for atherosclerosis and Alzheimer’s disease due to their importance in cholesterol metabolism [179,180].

In recent years, SOAT1 has started to gain significant consideration in the cancer field as accumulating evidence suggests that SOAT1 is a novel and promising drug target in many cancer types. Yue et al. reported that inhibition of cholesterol esterification by the SOAT1 inhibitors Avasimibe and Sandoz 58-035 suppresses prostate cancer cell proliferation and tumor growth in vivo [181]. Other groups also reported that SOAT1 is upregulated in prostate cancer tissues and is associated with earlier recurrence [182,183], and genetic inhibition of SOAT1 leads to the suppression of tumor growth in mice [182]. In hepatocellular carcinoma, SOAT1 is highly expressed in tumor tissues, and inhibition of SOAT1 by Avasimibe significantly inhibits tumor growth in vivo [184,185,186,187]. SOAT1 has also been shown to play an important role in pancreatic, colorectal, gastric, and lung cancers [188,189,190,191]. A novel SOAT1 inhibitor, Nevanimibe, also known as ATR-101, has been tested in a Phase I study in advanced adrenocortical carcinoma, although it showed only limited efficacy (Table 1) [192].

Our group discovered that SOAT1 expression is highly upregulated in tumor tissues from GBM patients, while SOAT2 is undetectable in tumor tissues [19,20]. Genetic or pharmacological inhibition of SOAT1 suppresses cholesterol storage and LD/CE formation, causing an increase in ER cholesterol, which suppresses SREBP-1 activation and reduces fatty acid synthesis, leading to suppression of GBM growth in vitro and in an orthotopic mouse model (Figure 2) [19,20]. As development of clinically effective pharmacological SREBP-1 inhibitors has failed in the past decades [193], indirect inhibition of SREBP-1 such as inhibition of SOAT1 may be an alternative strategy. Recently, high expression of SOAT1 was also found in glioma-associated macrophages, suggesting that inhibition of SOAT1 might also trigger an immune response to GBM cells [194]. One of the current challenges for SOAT1 inhibition in the clinic is that most SOAT1 inhibitors were manufactured as candidate drugs to treat atherosclerosis; thus, it is still unknown whether they could cross the blood–brain barrier (BBB). Therefore, development of novel SOAT1 inhibitors that could efficiently cross the BBB are worth developing and testing.

### 5.2. Targeting DGAT1 to Disrupt Fatty Acid Homeostasis for GBM Therapy

Fatty acids (FAs) are not only essential components of the membrane phospholipids but also serve as energy source through mitochondria-mediated β-oxidation [195]. Nevertheless, excess free fatty acid can impair cellular homeostasis and disrupts tissue function [107]. In 2003, Listenberger et al. reported that excess accumulation of palmitic acid (C16:0 saturated fatty acid) induces apoptosis in cultured CHO cells, while oleic acid (C18:1 monounsaturated fatty acid) supplementation leads to TAG accumulation [196]. Moreover, oleic acid rescues palmitate-induced apoptosis by channeling palmitate into TG pools, away from pathways leading to apoptosis. They suggested that palmitate-induced apoptosis in CHO cells occurs though a mechanism involving an increase in reactive oxygen species (ROS) and ceramide. The same year, another group reported that saturated fatty acids induce apoptosis in breast cancer cells through triggering cardiolipin turnover in mitochondria and cytochrome *c* release [197]. Interestingly, other studies reported that ceramide-mediated apoptosis could be prevented by generating acyl-ceramide and sequestrating it in LDs [198,199].

DGATs convert excessive free FAs into TAG that are stored in LDs [17,22]. In humans, there are two DGAT enzymes, DGAT1 and DGAT2, which are members of two different gene families and differ in their tissue distribution [200,201]. DGAT1 is highly expressed in the small intestine and DGAT2 is mainly expressed in liver and adipose tissues [202,203]. Loss of function mutations in DGAT1 were identified in patients with congenital diarrheal disorders [204] and protein-losing enteropathy [205]. DGAT2 knockout mice died soon after birth [206], while DGAT1 knockout mice only showed less TAG in liver and mammary glands [207]. Both DGAT1 and DGAT2 are located in the ER membrane and DGAT2 is a class I LD protein [109]. The two enzymes also have non-canonical acyltransferase activities with different substrate preference [201].

Our recent study demonstrated that targeting DGAT1 to block TAG synthesis causes severe FA metabolism dysregulation in GBM cells, leading to mitochondria damage and ROS production that trigger tumor cell apoptosis (Figure 3) [17,22]. We found that DGAT1 is overexpressed in tumor tissues from GBM patients and genetic or pharmacological inhibition of DGAT1, but not DGAT2, suppresses LD/TAG formation and reduces GBM tumor growth (Figure 3) [17]. Interestingly, inhibition of DGAT1 and DGAT2 is required to block LD/TAG formation and induce cell death in HepG2 cells [17]. It is still unclear why DGAT1 and DGAT2 behave differently in various cell types and which transcription factor regulates their expression in cancer cells. In addition, our study showed that the DGAT1 inhibitor A922500 dramatically suppresses GBM tumor growth in a subcutaneous mouse model (Table 1) [17]. Thus, it will be worthwhile to test if it effectively crosses the BBB. Targeting DGAT1 to block FA storage and disrupt homeostasis emerges as a promising new direction for GBM therapy. Furthermore, many publications have shown that DGAT1 is a better target than DGAT2 for cancer therapy [157,208,209,210,211,212,213].

## 6. Other Targets in LDs

Gene expression analysis shows that hypoxia-inducible lipid droplet associated (HILPDA), one of the LD-associated proteins, is highly expressed in bladder, colon, kidney, lung, pancreas, and uterine cancers as well as in GBM [214]. HILPDA increases LD accumulation by inhibiting TG hydrolysis by the adipose triglyceride lipase (ATGL) and stimulating TG synthesis via DGAT1 [210,215]. Mao et al. reported that HILPDA overexpression induced by hypoxia contributed to Bevacizumab resistance in GBM [216]. It may be a potential novel therapeutic target for GBM. Analysis of The Cancer Genome Atlas (TCGA) database shows that CGI-58, a co-activator of ATGL that stimulates TAG lipolysis and decreases LD numbers [217], is overexpressed in GBM and cervical cancer. Shi et al. reported that CGI-58 is upregulated in endometrial cancer and high CGI-58 expression was a poor prognostic marker for endometrial cancer [218]. However, a recent study showed that CGI-58 is a tumor suppressor, inducing cell cycle arrest in the G1 phase and causing growth retardation in a panel of prostate cancer cells [219]. The function of CGI-58 in GBM and whether it can affect tumor growth in GBM need to be explored further.

## 7. Conclusions and Future Directions

It is only recently that more scientists have realized the importance of lipid metabolism reprogramming in cancer cells. Our recent reports unveiled that lipid rewiring is a hallmark of GBM [1,22]. Drugs targeting lipid metabolism pathways such as de novo synthesis, uptake, storage, and catabolism are now being actively developed for cancer therapy, including therapy based on LD formation and regulation. Resolving the structures of SOAT1 [220,221,222] and DGAT1 [223,224] certainly will accelerate the discovery of potent inhibitors targeting these two enzymes for GBM therapy. Another alternative option is to develop new brain-targeted drug delivery systems based on nanoparticles to transfer effective drugs to the brain, possibly reducing the unexpected side effects due to offsite targeting.

Many questions remain regarding lipid metabolism regulation in GBM. Which transcription factors regulate the expression of SOAT1 and DGAT1? How GBM cells balance lipid synthesis, lipid storage, and lipolysis? Could potent SREBP-1 inhibitors be developed and tested clinically for cancer therapy? Are LDs playing a role in therapy resistance and tumor recurrence? Are there other targets regulating the formation of LDs in GBM as there are over 100 proteins locating or interacting with LDs [112,150]? Could other proteins in addition to DGAT1 and SOAT1 serve as druggable targets for cancer therapy? Addressing these questions certainly will provide new insights into the understanding of lipid metabolism in GBM and will help to develop novel therapeutic approaches. 

**Table 1 biomedicines-10-01943-t001:** Molecular targets and inhibitors in the lipid metabolism pathway for cancer therapy.

Target	Inhibitors	Types of Cancer	Preclinical Evidence	Clinical Trials	References
SREBPs	Fatostatin, Betulin, PF-429242, dipyridamole	GBM, breast, uterine, prostate, liver, lung, kidney, pancreatic and colon	Xenografts		[1,19,225,226,227,228,229,230,231,232,233,234,235]
ACLY	SB-201076, Hydroxycitric acid, Bempedoic acid, BMS-303141, SB-204990,	Lung, prostate	Xenografts		[1,236,237]
ACCs	ND-646, ND-654, soraphen A, TOFA	Lung, liver, breast	Xenografts, DEN-injured rats		[17,156,238,239,240,241,242,243,244]
FASN	TVB-2640, C75, cerulenin, orlistat	Lung, colon, breast, GBM, astrocytoma, prostate, uterine	Xenografts	NCT03808558 NCT02223247 NCT02980029 NCT03179904 NCT05118776 NCT03032484	[185,187,192,245,246,247,248,249]
SCD1	A939572, MF-438, CAY10566 MK-8245	Lung, pancreatic, kidney, GBM, ovarian	Xenograft, Pdx1Cre; LSL-KrasG12D mouse	NCT00972322 (type 2 diabetes)	[164,250,251,252,253,254,255,256]
ACSS2	Tetrazoles, Pyridine Derivatives				[257,258]
SLC25A1	CTPi2, BTA, CNASB				[259,260]
CPT1A	Etomoxir, Perhexiline, ST1326	GBM, breast, leukemia, prostate, colon	Xenografts, Transgenic mice		[261,262,263,264,265,266,267,268]
DGAT1	A922500, AZD3988, PF-04620110	GBM, liver, uterine, prostate	Xenografts		[17,156,242,243,244,269]
SOAT1	Avasimibe, ATR-101, K604,	Liver, colon, GBM, kidney	P53-deficient mice, AOM/DSS-treated and ApcMin/+ mice, xenografts	NCT01898715	[185,187,192,246,247]

## Figures and Tables

**Figure 1 biomedicines-10-01943-f001:**
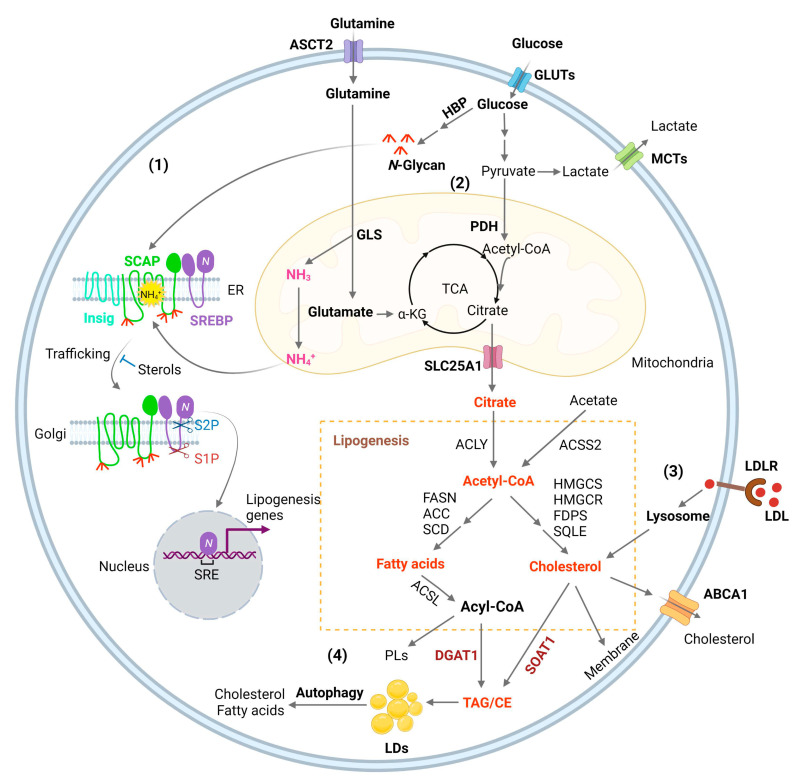
Lipid metabolism reprogramming in cancer cells. (**1**) Combined elevation of glucose and glutamine consumption promotes lipogenesis by activating the SREBP/SCAP pathway. Glucose produces N-glycans through HBP (Hexosamine Biosynthesis Pathway), which stabilizes SCAP (SREBP-cleavage activating protein) through N-linked glycosylation of its luminal loops (1 and 7). Glutamine enters the mitochondria and ammonia (NH3) is released by GLS (glutaminase). NH3 is protonated and converted to NH4+, which directly binds to the aspartate D428 and Serine S326/330 in the core of the SCAP transmembrane domains, forming stable hydrogen bonds. This binding triggers a dramatic conformation change in SCAP, leading to its dissociation from Insig (insulin-induced gene), an endoplasmic reticulum (ER)-resident protein. Subsequently, SCAP escorts SREBP (sterol regulatory element-binding protein) to the Golgi, where it is cleaved by two enzymes S1P (site 1 protease) and S2P (site 2 protease) to release its active N-terminal fragment. Finally, the N-terminal domain goes into the nucleus, binds to the SRE (sterol regulatory element) motif located in the promoters of gene involved in lipogenesis to activate their transcription and promote de novo lipid synthesis and tumor growth. (**2**) Glucose is the main source for lipid synthesis. Glucose via glycolysis breaks down into pyruvate, which enters the mitochondria and is converted to acetyl-CoA by PDH (pyruvate dehydrogenase), followed by condensation with OAA (oxaloacetate) to form citrate to enter the TCA cycle (tricarboxylic acid cycle). Citrate is released to the cytosol via its mitochondria transporter, SLC25A1. Citrate is then cleaved to acetyl-CoA by ACLY (ATP citrate lyase), which serves as a precursor for fatty acid and cholesterol biosynthesis catalyzed by a series of enzymes that are the main transcriptional targets of SREBPs, as shown in the Figure. In addition, cytosol acetate can be converted to acetyl-CoA for lipid synthesis by ACCS2 (acetyl-CoA synthetase 2). Besides that, acetyl-CoA is also the substrate for the acetylation of histones, which is involved in epigenetic regulation. Moreover, glutamine through glutaminolysis contributes as an anaplerotic substrate to replenish tricarboxylic acid (TCA) cycle intermediates. Glutamate, the product of the first step of glutaminolysis, is converted to α-KG (α-ketoglutaric acid) and then enters into the TCA cycle. (**3**) SREBPs upregulates the expression of LDLR (low-density lipoprotein receptor), which binds to LDL and transports it into cells through the endocytosis process. LDL is then hydrolyzed in the lysosomes and cholesterol is released, promoting tumor growth. (**4**) Excess fatty acids and cholesterol are converted to TAG (triacylglycerol) and CE (cholesteryl ester) by DGAT1 (diacylglycerol O-acyltransferase 1) and sterol O acyl-transferase 1 (SOAT1) to form LDs (lipid droplets) and prevent toxicity from high lipid levels. Under conditions of nutrient deficiency, LDs are hydrolyzed by autophagy to release free fatty acids and cholesterol for tumor survival.

**Figure 2 biomedicines-10-01943-f002:**
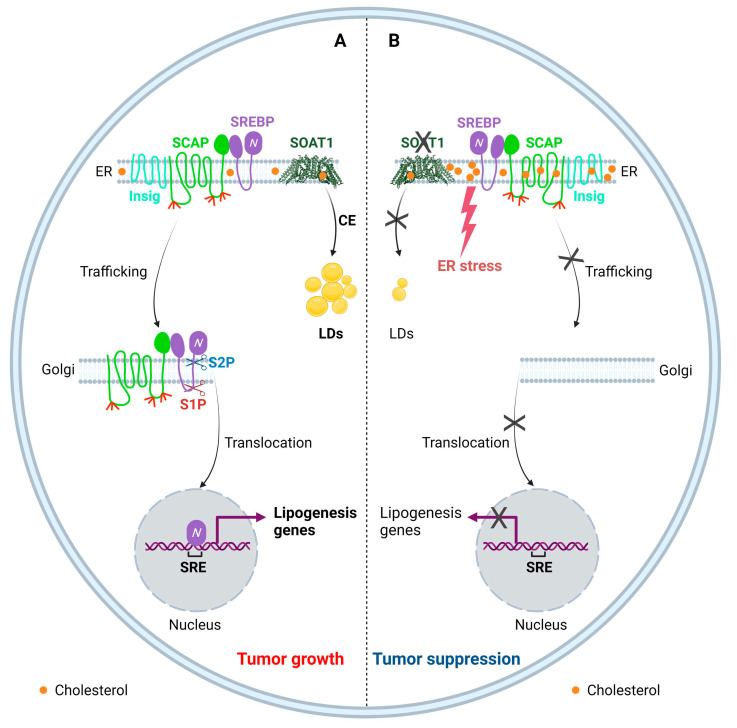
Targeting SOAT1 to disrupt cholesterol homeostasis for GBM therapy. (**A**) SOAT1 catalyzes the esterification of cholesterol to store excess free cholesterol into LDs, and thus is involved in maintaining cellular cholesterol homeostasis. Cholesterol negatively regulates SCAP/SREBP trafficking and SREBP activation. Cholesterol reduction in the ER activates SCAP dissociation from Insig, resulting in SREBP translocation and activation to promote lipogenesis. (**B**) Inhibition of SOAT1 reduces CE formation, which in turn results in the accumulation of cholesterol in the ER membrane, subsequently leading to the SCAP/SREBP complex remaining in the ER and to reduction of lipogenesis, along with enhanced ER stress, resulting in tumor suppression.

**Figure 3 biomedicines-10-01943-f003:**
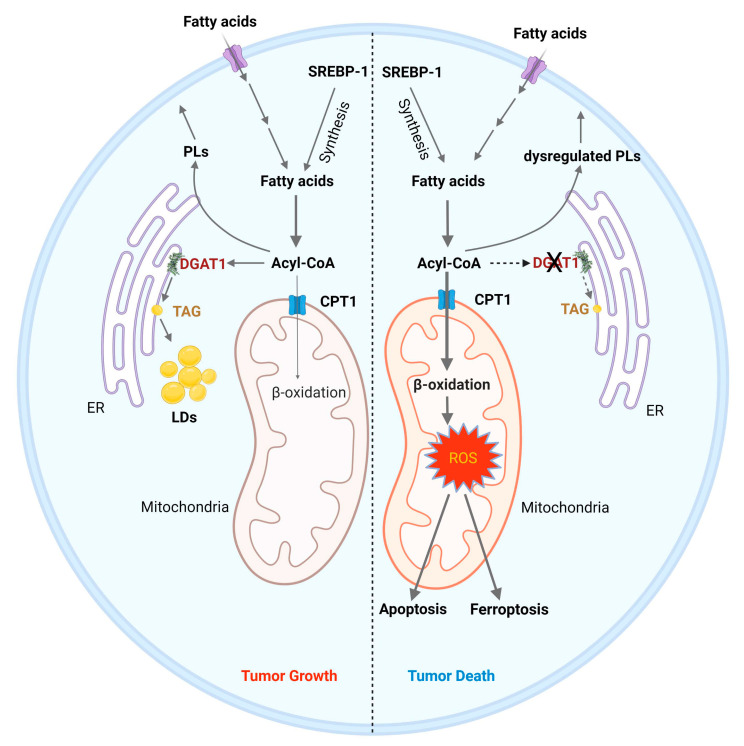
Inhibition of DGAT1 as a therapy strategy to induce cancer cell death. (1) **Left**: Tumor cells acquire fatty acids through uptake or de novo synthesis via the activation of SREBP-1. Fatty acids are converted to acyl-CoAs, which are the substrates for phospholipid (PL) synthesis and produce energy via entering into mitochondria to undergo β-oxidation and oxidative phosphorylation. Excessive acyl-CoA is stored in LDs as catalyzed by DGAT1 in tumor cells. (2) **Right**: Inhibiting DGAT1 causes an imbalance of fatty acid catabolism, leading to cell death. Inhibiting DGAT1 forces more acyl-CoAs to enter the mitochondria through carnitine palmitate transferase 1 (CPT1) for β-oxidation, leading to high levels of reactive oxygen species (ROS), which trigger apoptosis and ferroptosis to kill tumor cells. Moreover, the profile of PLs is altered upon DGAT1 inhibition, contributing to further cellular stress.

## Data Availability

Not applicable.

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
