# Peer review of "Lipid Metabolism in Glioblastoma: From De Novo Synthesis to Storage"

_biomedicines, 2022, doi:10.3390/biomedicines10081943_

Round 1

Reviewer 1 Report

Great paper and interesting topic. It's very well organized and provides a good overview of lipids in glioma. Very well written.

Author Response

Thanks very much for Reviewer 1 to spend precious time reviewing our manuscript and support our work for publication. 

Reviewer 2 Report

This manuscript shows a good overview about the lipid metabolism in glioblastoma. In general it well described and written. However, minor revisions are needed:

- Second paragraph: line 8: In the text you mention that "our recent study demonstrated that GBM cells prevent lipotoxicity....", however, I would suggest to add "our recent studies".

- Authors declared in the conclusions and future directions section "Resolving the structures of SOAT1 and DGAT1 certainly will accelerate the discovery of BBB-crossing inhibitors for GBM therapy". Could you extend this afirmation? 

- Authors should consider adding the model of their studies (if their results are shown in cell models, animal models, patients' samples...)

- Authours should consider adding a list of abbrevetions

Author Response

Thanks very much for Reviewer #2 for reviewing our manuscript and great advice. Below is the point-by-point response to the Reviewer's comments. 

- Second paragraph: line 8: In the text you mention that "our recent study demonstrated that GBM cells prevent lipotoxicity....", however, I would suggest to add "our recent studies".We changed "study" to "studies".

-Authors declared in the conclusions and future directions section "Resolving the structures of SOAT1 and DGAT1 certainly will accelerate the discovery of BBB-crossing inhibitors for GBM therapy". Could you extend this afirmation? We modified the sentence to "Resolving the structures of SOAT1 [221-223] and DGAT1 [224, 225] certainly will accelerate the discovery of potent inhibitors targeting these two enzymes for GBM therapy. The previous statement is overstated. 

-Authors should consider adding the model of their studies (if their results are shown in cell models, animal models, patients' samples...)Yes, all the models were added in related sections. 

-Authors should consider adding a list of abbrevetions

Abbreviation were added. 

Round 2

Reviewer 1 Report

Well written paper, I have no concerns. It is a very good synopsis on lipid metabolism in GBM. I think this is a great paper.

Reviewer 2 Report

First of all I would like to thank the authors for all the changes that have incorporated to the manuscript.

However, I would like to point out that when they have added "our recent studies" they have include "studiesy". Please check the word and correct it.

Also, in the section 3.1 paragraph 2, please verify that in line 2 after "lactating breast, and  ...." there is only one space.

Thank you again and congratulations